# Role of Ultrasound Evaluation of Temporomandibular Joint in Juvenile Idiopathic Arthritis: A Systematic Review

**DOI:** 10.3390/children9081254

**Published:** 2022-08-19

**Authors:** Achille Marino, Orazio De Lucia, Roberto Caporali

**Affiliations:** 1Unit of Pediatric Rheumatology, ASST G. Pini-CTO, Via Gaetano Pini 9, 20122 Milan, Italy; 2Division of Clinical Rheumatology, ASST G. Pini-CTO, Via Gaetano Pini 9, 20122 Milan, Italy; 3Department of Clinical Sciences and Community Health, Research Center for Pediatric and Adult Rheumatic Diseases (RECAP.RD), University of Milan, Via della Commenda 19, 20122 Milan, Italy

**Keywords:** juvenile idiopathic arthritis, temporomandibular joint, synovitis, ultrasonography

## Abstract

**Background:** Juvenile idiopathic arthritis (JIA) is childhood’s most frequent chronic rheumatic disease. JIA is a broad term that includes all arthritides starting before 16 years, lasting at least six weeks, and of unknown cause. The temporomandibular joint (TMJ) could be involved in JIA both at onset and during the disease course. The presence of TMJ synovitis might severely impair dentofacial maturation in pediatric patients. The ultrasound (US) application to detect early signs of TMJ synovitis in children with JIA has provided contradictory results. We sought to assess the current role of TMJ US in JIA through a systematic literature review. **Methods:** The systematic review was conducted according to the recommendations of the Preferred Reporting Items for Systematic Review and Meta-Analysis (PRISMA). **Results:** The literature search found 345 records. After duplicates removal, 253 records were screened, 20 full-text articles were reviewed to assess their eligibility, and 7 of them were included in the qualitative analysis. Joint effusion was the most recorded parameter, followed by bony condylar abnormalities. Compared to contrast enhancement MRI, the capability to detect signs of active synovitis of TMJ by US is low, especially at the early stages. **Conclusion:** Understanding how US may help diagnose and manage children with JIA is advisable for several reasons. MRI cannot be frequently repeated, may need sedation, and is expensive. The constant technical improvement of US will undoubtedly allow for better evaluation of what, in the past, was not clear or not even captured by sonography. So far, the role of US in the assessment of TMJ involvement in JIA is indubitably secondary to the MRI. Even so, we think that a baseline MRI of TMJ and the repetition of the sonography over time might both help the interpretation of US images and intercept significative changes.

## 1. Introduction

Juvenile idiopathic arthritis (JIA) is childhood’s most frequent chronic rheumatic disease, with a prevalence estimated between 16 and 150 per 100,000. JIA is a broad term that includes all arthritides starting before 16 years, lasting at least six weeks, and of unknown cause [1,2]. According to specific characteristics displayed in the first six months of the disease, the International League of Associations for Rheumatology (ILAR) recognizes seven main categories: oligoarticular JIA, polyarticular rheumatoid factor (RF) negative JIA, polyarticular RF positive JIA, psoriatic arthritis, enthesitis-related arthritis, systemic JIA, undifferentiated JIA [3]. Nevertheless, efforts are currently ongoing to develop a new classification of inflammatory arthritis with particular regard to biological phenotypes [4]. 

The temporomandibular joint (TMJ) could be involved in JIA both at onset and during the disease course; TMJ arthritis may affect a substantial number of JIA patients (up to 87%) [5,6,7]. The presence of TMJ synovitis might severely impair dentofacial maturation in pediatric patients. The condyle is the principal center of mandibular growth; the cartilage ossification at this level causes an increase in the vertical height of the ramus, and puberty accelerates mandibular growth, which usually ends by the age of 18 years [8,9]. Indeed, the growth site lies on the mandibular condyle in the intracapsular joint space and can be easily damaged by arthritis [10]. 

In the general population, the causes of temporomandibular disorders (TMD) are various; indeed, several factors may influence the development of this alteration: traumatic or post-traumatic abnormalities, intrinsic anatomic factors, and systemic conditions [11].

In JIA patients, the term dentofacial deformity refers to the alteration in facial bone growth, development, and structure due to TMJ arthritis [12]. Micrognathia, retrognathia, and frontal facial asymmetry are the most frequently reported dentofacial deformities in JIA patients, as well as the deviation of the mandible on maximal mouth opening [13,14,15,16].

Therefore, the early recognition and adequate treatment of TMJ arthritis are advocated. Unfortunately, the assessment of TMJ involvement in JIA is still challenging. Indeed, signs of TMJ synovitis can be absent at onset, and patients might develop symptoms afterward when the damage is already present. Therefore, clinical examination is not enough to exclude active arthritis, nor to rule out other diagnoses [17,18]. In this setting, imaging plays a crucial role in the diagnosis of TMJ involvement, both for acute arthritis and residual damage. Gadolinium-enhanced MRI (Gd-MRI) represents the gold standard for the diagnosis of active synovitis of TMJ in JIA patients, even for early stages, and efforts are still ongoing to develop a validated TMJ-MRI scoring system [19,20]. 

Nevertheless, the high cost, the exam length, and the frequent need for sedation are considered significant disadvantages of this imaging modality, limiting its application in daily clinical practice. Furthermore, doubts have been raised regarding the sensitivity of an abnormal TMJ-MRI exam. Indeed, a slight effusion (<1 mm of diameter) and/or contrast enhancement, in the absence of chronic changes, has been reported in healthy children as well [21]; therefore, a careful interpretation of Gd-MRI images is always advisable. Ultrasound (US) represents a rather appealing way to look at the joints since it is inexpensive, fast, noninvasive, free of ionizing radiation, and can be frequently repeated. However, US does need an expert sonographer; from a technical point of view, it cannot penetrate through both bones and dense, soft tissue structures. The US application to detect early signs of TMJ synovitis in children with JIA has provided contradictory results [22,23,24,25,26,27,28,29]. Figure 1 shows US images of TMJ. We sought to assess the current role of TMJ US in JIA through a systematic literature review. The PICO table (population, intervention, comparison, and outcome) is herein reported (Table 1).

## 2. Materials and Methods

A systematic review was conducted according to the recommendations of the Preferred Reporting Items for Systematic Review and Meta-Analysis (PRISMA) [30]. This review aimed to report the current role of US in the assessment of TMJ arthritis in children with JIA. 

### 2.1. Search Strategy

A literature search was developed first for a MEDLINE search and then adapted, if needed, to comply with the syntaxes of the following databases: EMBASE, WEB OF SCIENCE, and SCOPUS. The following keywords were used for the search: “juvenile idiopathic arthritis” OR “juvenile rheumatoid arthritis” OR “juvenile chronic arthritis” OR “juvenile idiopathic arthritides” OR “juvenile chronic arthritis” OR “Still disease” OR “still disease” OR JIA OR JCA OR JRA AND “temporomandibular joint” OR “temporomandibular joints” OR TMJ AND “Ultrasound” OR “Ultrasonography” OR “Sonography” OR “US”. The literature search was conducted through March 2022. 

### 2.2. Inclusion/Exclusion Criteria

The inclusion criteria were randomized clinical trials, cohort studies, observational studies, and case series using ultrasound for TMJ involvement assessment in JIA patients. The exclusion criteria were the following: insufficient data regarding US use; lack of comparison indicators (either TMJ-MRI or healthy controls); non-peer-reviewed studies (e.g., conference abstracts); articles mainly reporting on JIA patients older than 20 years of age; case series reporting fewer than five JIA patients; and non-English-written studies. 

### 2.3. Study Selection and Quality Assessment

Two reviewers (AM and ODL) reviewed the results independently. Duplicates were removed. The first evaluation relied on titles only, and then abstracts of the included titles were screened. Results considered relevant were reviewed in full text and evaluated if they met selection criteria. Two authors (AM and ODL) manually evaluated the references of each full paper assessed. In case of disagreement, a third reviewer (RC) helped to make the final decision. A discussion of the original article was conducted, and if no accord could be reached, RC was then involved. Bias criteria were adapted according to the low level of evidence within this field using Cochrane guidelines as reference [31,32]. Table 2 shows the evaluated items. 

## 3. Results

The literature search found 345 records: 183 from Medline, 28 from Embase, 86 from Web of Science, and 48 from Scopus. Nine articles were added from the reference list of records considered interesting. After duplicate removal, 253 records were screened, and 20 full-text articles were reviewed to assess their eligibility [22,23,24,25,26,27,28,29,33,34,35,36,37,38,39,40,41,42,43,44]. In total, 13 of them were excluded: 10 because they did not report enough US data, and 3 because of the lack of comparison indicators. Therefore, seven records were finally assessed for qualitative analysis (Table 3).

Figure 2 shows a PRISMA diagram. All were prospective studies; one had no TMJ-MRI done but had a control group of healthy age- and sex-matched children [25], and one single study had both TMJ-MRI assessment and a control group [29]. The median number of included subjects was 32, ranging between 8 and 92 patients. The age of the enrolled patients was comparable among the studies. The US scan was performed by a radiologist in all but one study, in which a rheumatologist performed the procedure [25]. Joint effusion was the most recorded parameter (six studies), followed by bony condylar abnormalities (four studies). The only other two parameters evaluated were synovial thickening (two studies) and power doppler signal (one study). Joint effusion and bony condylar abnormalities were recorded in four studies [23,24,25,26], with one of them also analyzing the simultaneous presence of synovial thickening [26]. One study analyzed the presence of joint effusion and synovial thickening [29]. Conversely, two studies only analyzed one parameter each (power doppler signal and joint effusion, respectively) [27,28]. Given the heterogeneity in the studies, no meta-analysis has been performed. 

### 3.1. Joint Effusion

Joint effusion was clearly defined in three out of the six studies in which this parameter was evaluated: capsular width was used in three studies, but with different cut-offs [25,27].

Weiss et al. defined acute arthritis as the presence of joint effusion, but they did not observe any joint effusion among the 24/32 patients of their cohort who showed acute TMJ arthritis at MRI [23].

Muller et al., using a capsular cut-off of 2 mm, detected the presence of joint effusion in 8/29 (28%) patients and 10/58 (17%) joints. At the same time, signs of active TMJ synovitis were documented in 19/30 patients (63%) and 32/60 joints (53%) at MRI [24]. Melchiorre et al. reported a high rate of joint effusion (capsular cut-off of 1.5 mm) among newly diagnosed JIA patients (68%). Unfortunately, no TMJ-MRI assessment was performed, but no joint effusion was detected among the 40 healthy controls [25]. In the study by Assaf et al., US images of 20 patients with TMJ involvement previously diagnosed by MRI were analyzed. Four images of each TMJ were evaluated. Joint effusion was detected in 20 out of 160 images. Unfortunately, study results were reported in terms of positive images only, making abstracting patients’ rates impossible [26]. Kirkhus et al. compared MRI findings with US scans in 55 JIA patients; the capsular width was measured both at condylar and subcondylar levels and compared with different degrees of synovitis documented by MRI. The ROC curve showed a better performance when capsular width was measured at the subcondylar level with a proposed cut-off of 1.2 mm, a sensitivity of 72%, and a specificity of 70% [27]. Tonni et al. analyzed US findings of eight JIA patients with MRI confirmed TMJ involvement and seven healthy controls. They did not detect joint effusion by US either in the JIA patients or in the healthy controls [29]. 

### 3.2. Condylar Changes

In the study by Weiss et al., chronic TMJ arthritis, defined as the presence of condylar changes and/or erosions, was found in 9 patients (28%) by US and in 22 (69%) by MRI [23]. 

Deformities of the mandibular condyle (surface irregularity, erosion, flattening, and acute angulation at the transition point from the lateral to the superior condylar surface) were detected in 7/29 (24%) patients by US and in 9/30 (30%) in the study by Muller et al. [24]. Melchiorre et al. observed the presence of erosion, osteophyte formation, and bone remodeling. Condylar remodeling was observed in 62/68 patients, with bilateral involvement in the majority of them (37 cases). Erosions were found in 18 out of 124 TMJs, while 14/124 TMJs had osteophytes [25]. In the study by Assaf et al., surface irregularities and erosions were observed in 40 and 124 out of 160 images, respectively. In the same study, the thickness of the condylar disc was also evaluated (cut-off value of 1.57 mm), with 48 abnormal images out of 160 being recorded [26]. 

### 3.3. Synovial Thickening

Assaf et al. assessed the synovial thickening using a cut-off value of 1.56 mm. They found 55/160 abnormal images that documented a concurrent narrowing of the joint space [26].

Tonni et al. used the lateral periarticular joint space (LPAS) measured from the condyle’s cortical contour to the capsule’s contour as indirect parameters of the synovial joint space [29]. The measures of LPAS were significantly different between the JIA patients and the controls (0.086 cm and 0.0055 cm, respectively; *p* = 0.000); furthermore, the coronal closed-mouth position resulted in the most appropriate acquisition mode, given the narrowest confidence interval (0.022). However, the authors stated that LPAS does not help differentiate acute from chronic synovitis [29]. 

### 3.4. Power Doppler Signal

Ultrasound can document synovial vascularization and hyperemia through a power Doppler signal; this amplifies the US sensitivity for acute inflammation in several joints [45]. Only the study by Zwir et al. evaluated this parameter. In this study, 92 consecutive JIA patients were enrolled; all the patients had power Doppler US and MRI performed on the same day [28]. No power Doppler signal was documented among the study cohort. On the other hand, MRI detected contrast enhancement in 119 joints out of 184 TMJs. 

### 3.5. Comparison Indicators

In all except one study [25], TMJ-MRI was performed; however, Assaf et al. used this imaging modality as the entry criterion (all the included patients had TMJ involvement detected by MRI without any further analysis) [26]. Weiss et al. compared US and MRI for acute and chronic changes, suggesting a poor correlation, especially for acute arthritis. They found a 23% agreement for acute arthritis (kappa value of 0) and a 50% agreement for chronic changes (kappa value of 0.12) [23]. On the other hand, Muller et al. reported a statistically significant correlation with active TMJ arthritis on MRI (chi-square *p* 0.047 for patients). Nevertheless, the level of the contrast enhancement at MRI influenced the correspondence between US and MRI: an 88% agreement in severe enhancement compared to 33% in moderate enhancement (chi-square *p* 0.003 for the difference) [24]. Kirkhus et al. found a moderate correlation in capsular width detected by US and synovitis at MRI (Spearman’s rho: 0.483 at the subcondylar level *p* = 0.001 and 0.347; *p* = 0.001 at condylar level) [27]. In comparison with MRI, the power Doppler signal alone, as analyzed by Zwir et al., showed poor sensitivity and specificity (0% and 36.4%, respectively) for TMJ inflammation [28]. The values of LPAS measured with US and MRI showed a positive correlation in the study by Tonni et al. (Spearman test: *p* of 0.623 and a *p* < 0.05) [29].

In comparison with the healthy controls, low values of TMJ capsule width (<1.4 mm) were found in the 40 controls by Melchiorre et al. [25]. In the other study with a comparison cohort, Tonni et al. reported a lower value of LPAS in healthy children than in JIA patients (0.0055 cm vs. 0.086 cm) [29]. 

## 4. Discussion

The technological improvements and accessibility, the low cost, the time-consumption advantages, and the lack of ionizing radiations all represent the main reasons for the spread of daily US usage in rheumatology. With regard to pediatric rheumatology, interpreting US images of JIA patients is particularly challenging given the changes in children’s growth. Nevertheless, the role of US as a diagnostic tool in JIA has not been exhaustively established yet [46]. Sonographers’ experience and advanced technology are crucial in avoiding misleading interpretations of US images. Furthermore, efforts have been made to standardize the physiological appearance of certain joints at US (but not TMJ) [47,48,49]. 

The peculiarity of the TMJ makes it hard to evaluate by US, since only the lateral aspect of this joint can be seen with this imaging modality [50]. This justifies the paucity and the low quality of studies on US as a diagnostic tool for TMJ synovitis in JIA. Furthermore, the different methodologies used and the US parameters analyzed make the herein retrieved studies challenging to compare. A timespan of more than ten years among the studies should be taken into consideration, especially when technical aspects might have changed over the years (machine brands, probe position descriptions, and frequency of linear transducers). All except one study used MRI for reference. However, there was no uniformity in the definition of active arthritis by MRI. For example, unlike other authors, Weiss et al. did not consider contrast enhancement as a sign of active inflammation. In this regard, efforts to develop a standardized TMJ-MRI scoring system are ongoing [20,51,52]. 

It is appropriate to highlight how a standardized image acquisition protocol for TMJ in children with JIA is necessary. Indeed, the difficulty in comparing the studies lies principally in the different parameters analyzed and their respective definitions. Even so, several considerations can be abstracted. No study demonstrated the superiority or the equivalence of US to MRI, which remains by far the best imaging modality to detect early signs of TMJ arthritis. Furthermore, a baseline MRI might be helpful for the US image interpretations. Indeed, US accuracy may be improved in light of the MRI result, as documented by a non-JIA study [53].

By analyzing the studies collected herein, the weakest point of US seems to be the capability to detect signs of active synovitis, especially at the early stages. This can partially be explained by the peculiarity of TMJ, both anatomically and functionally: a little joint set among the temporal bone and the mandible, the presence of the articular disc separating the superior from the inferior part, and the possibility of both sliding and hinge movements (ginglymoarthrodial joint). The need for synchronism with the contralateral TMJ represents a singularity that further complicates its evaluation. By the current evidence reported in this study, joint effusion does not appear to be a suitable parameter of active TMJ arthritis; nor does the analysis of the power Doppler signal. Synovial thickening seems more sensible than the abovementioned parameters. However, it is interesting to note that no study evaluated joint effusion, synovial thickness, and power Doppler signal altogether, despite these parameters being normally assessed in a routine US for evaluating the other joints. Furthermore, other specific parameters could help to characterize TMJ involvement in JIA. Disc dislocation detected by US has been found to play a role in the diagnosis of TMJ disorders; whether this is also true in JIA should be investigated [54]. 

Understanding how US may help diagnose and manage children with JIA is advisable for several reasons. MRI cannot be frequently repeated, may need sedation, and is expensive. Furthermore, knowledge about the gadolinium-based contrast medium effects is growing and, besides the nonallergic and allergy-like reactions, additional complications such as brain deposition are nowadays a matter of concern [46,47,48,49,50,51,52,53,54,55]. 

To date, few studies have analyzed the diagnostic value of US in TMJ synovitis of JIA patients. The parameters used in these studies were heterogeneous and needed clear definitions. The presence of joint effusion alone, the most frequently analyzed measure, has shown a modest correlation with MRI findings.

The constant technical improvement of US will undoubtedly allow for a better evaluation of what, in the past, was not clear or not even captured by sonography. As yet, the role of US in the assessment of TMJ involvement in JIA is indubitably secondary to the MRI. Even so, we think that a baseline MRI of TMJ and the repetition of the sonography over time might both help the interpretation of US images and intercept significative changes. Whether the US sensitivity for active synovitis shall be improved in the future, this paradigm could be inverted as US would become an exam of the first level for TMJ arthritis to select those patients who need an MRI.

The current efforts made for the standardized evaluation of TMJ-MRI could be essential to developing a US acquisition protocol. Indeed, using a shared MRI protocol as a reference may serve as a guide to understanding the more suitable parameters of normality and active synovitis by US. 

## Figures and Tables

**Figure 1 children-09-01254-f001:**
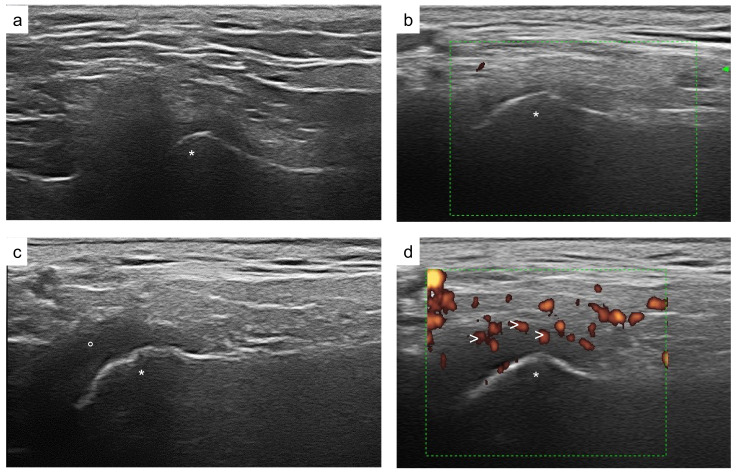
Ultrasound images of the temporomandibular joint. Regular TMJ aspect of grayscale and Power Doppler signal (**a**,**b**); abnormal TMJ findings: TMJ effusion (**c**); increased intrasynovial power doppler signal (**d**); * mandibular condyle; ° intraarticular synovial fluid; > intrasynovial power doppler signal.

**Figure 2 children-09-01254-f002:**
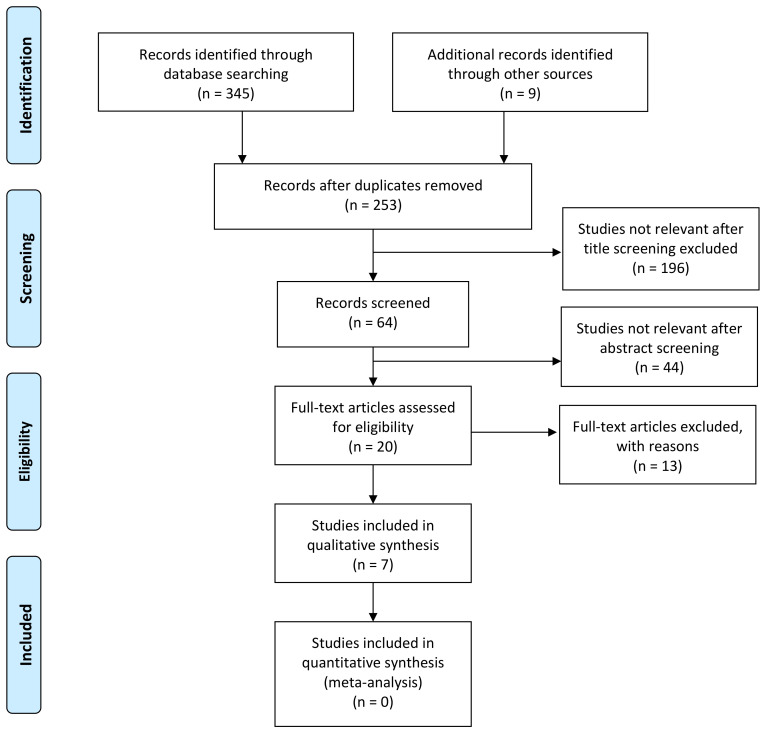
PRISMA flow diagram.

**Table 1 children-09-01254-t001:** The PICO table (population, intervention, comparison, and outcome).

*PICO*
***Patients:*** Patients with JIA evaluated for TMJ involvement ***Intervention:*** Use of US for assessment of TMJ involvement ***Comparison:*** Comparison with healthy controls and/or MRI for evaluation of TMJ involvement ***Outcome:*** Identify active synovitis of TMJ in JIA patients **Database included:** Medline via PubMed, EMBASE, WEB OF SCIENCE, SCOPUS

**Table 2 children-09-01254-t002:** The quality assessment of included studies.

Study	Study Type	Level of Evidence	Uniform Inclusion Criteria	Standardized Imaging Protocol	Sufficient Outcome Variables Description	Blinded Assessor	Risk of Bias
**Weiss** [23]	Prospective	3b	No	Yes	No	Unclear	High
**Muller** [24]	Prospective	4	Yes	Yes	Yes	Yes	Low
**Melchiorre** [25]	Prospective	4	No	Yes	Yes	Yes	High
**Assaf** [26]	Prospective	3b	No	Yes	Yes	No	High
**Kirkhus** [27]	Prospective	4	No	Yes	No	Yes	High
**Zwir** [28]	Prospective	4	Yes	Yes	No	Yes	High
**Tonni** [29]	Prospective	3b	No	Yes	Yes	Unclear	High

Level of evidence: Oxford Centre for Evidence-Based Medicine using the protocol for differential diagnosis/symptom-prevalence studies http://www.cebm.net/oxford-centre-evidence-based-medicine-levels-evidence-march-2009/ (accessed on 3 May 2022).

**Table 3 children-09-01254-t003:** Description of included studies.

	Weiss et al. (2008) [23]	Muller et al. (2009) [24]	Melchiorre et al. (2010) [25]	Assaf et al. (2015) [26]	Kirkhus et al. (2016) [27]	Zwir et al. (2020) [28]	Tonni et al. (2021) [29]
**N of JIA pts (female)/joints**	32 (25)/64	30(16)/60	68 (57)/136; 40 healthy controls	20 (16)/40	55 (42)	92(63)/184	8(7)/14; 7 healthy controls
**Age, years**	8.6 (median)	9.8 (median)	11 (mean)	11.06 (mean)	12.4 (mean)	12.7 (mean)	11.6 (mean)
**Transducer frequency**	12.5 MHz	12 MHz	8.5 MHz	12 MHz	12–18 MHz	13 and 6.7 ** MHz	15 MHz
**Who did the US?**	Radiologist	Radiologist	Rheumatologist	Radiologist	Radiologist	Radiologist	Radiologist
**Joint effusion**	0 pts (Defined as fluid collection in the joint)	8/29 (28%) pts and 10/58 (17%) joints (Defined as thickening of the joint capsule (>2 mm))	46/68 pts (68% (bilateral in 16 (35%) cases) (Defined as thickening of the joint capsule >1.5 mm and the presence of a hypoechoic area within the joint space)	20 positive images/160 (12.5%) (Defined as sonographically visible fluid accumulation within the articular space)	Sensitivity 72%, specificity 70% for the capsular width at the subcondylar level * (Capsular width was measured as an indirect measurement of synovitis. Capsular cut-off of 1.2 mmL)	NE	0 pts
**Synovial thickening**	NE	NE	NE	55 positive images/160 (34.4%) (Defined as a value greater than 1.56 mm)	NE	NE	LPAS of JIA pts 0.086 cm LPAS of controls 0.0055 cm (Evaluated as the lateral periarticular space (LPAS) Defined as the width of the synovial joint space measured from the cortical contour of the condyle to the contour of the capsule)
**Synovial PWD**	NE	NE	NE	NE	NE	0 pts	NE
**Condylar changes**	9 pts (28%)	7/29 (24%) pts and 10/58 (17%) joints	62 (91.2%) out of 68 pts	124 positive images/160 (77.5%)	NE	NE	NE
**TMJ-MRI assessment and/or healthy controls comparison**	23% agreement and a kappa coefficient of 0 for acute TMJ arthritis 50% agreement and a kappa coefficient of 0.12 for chronic TMJ involvement	A pathological US was statistically significantly correlated with active TMJ arthritis on MRI (chi-square *p* 0.008 for all joints and *p* 0.047 for patients)	No TMJ-MRI assessment. In all 40 healthy controls, the TMJ capsule was less than 1.4 mm thick	For every enrolled patient the involvement of the TMJ was proven by MRI	The correlation between ultrasonography-assessed capsular width and MRI-assessed amount of synovitis was moderate both at the subcondylar and at the condylar levels (Spearman’s rho (r): 0.483; *p*, 0.001 and 0.347; *p*, 0.001, respectively).	Poor sensitivity (0%), low specificity (36.4%), very low positive predictive value (0%), and high negative predictive value (100%) when compared with MR	The Spearman test applied to the values of LPAS measured in ultrasound and the corresponding MR images showed a proportional positive correlation with a *p* of 0.623 and a *p* < 0.05

* No number of positive findings was recorded. ** Multi-frequency linear probe at a maximum frequency of 13 MHz for gray-scale US and 6.7 MHz for power Doppler US. JIA: juvenile idiopathic arthritis; pts: patients; PWD: power Doppler; US: ultrasound; MRI: magnetic resonance imaging; TMJ: temporomandibular joint.

## Data Availability

Data are available on request to the authors.

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
