# Peer review of "Role of Ultrasound Evaluation of Temporomandibular Joint in Juvenile Idiopathic Arthritis: A Systematic Review"

_children, 2022, doi:10.3390/children9081254_

Round 1

Reviewer 1 Report

PRISMA flow chart is not visible in pdf file.

The article is written correctly. The subject is interesting, current  and important.  I have no comments.

Author Response

Author answer: Glad for your opinion. PRISMA flow chart is now visible. Thank you.

Reviewer 2 Report

Dear Authors,

1. The manuscript requires extensive editing of English language and style.

2. Abstract should be improved, especially background and conclusion.

3. Why do you mention classification by ILAR? You found this claasification unsatisfactory. Either describe this classification in details or remove it. Introduction is the part of work in which you present what is known about the subject and explain why you decided to write this review- avoid sharing your opinions in introduction.

4. Check carefully within the literature the current state of knowledge regarding the growth site in mandibular condyles.

5. Add typical characteristics (phenotype) of JIA patients regarding head morphology, including: retrgnathic profile, antegonial notch, symphysis, face.

6. Lines 51-52. Small effusion in TMJs can NEVER be seen in healthy children. 
7. PRISMA flow diagram is not visible.

8. What does the term „condylar changes” mean? What Kind of condylar changes were the authors looking for?

9. Explain what power Doppler signal is and what information it gives for the clinicians.

10. You mentioned that US is indubitably ancillary to MRI. In fact, you did not mention what additional information we get from US which we do not get from MRI. If there is nothing superior, US cannot be called: „indubitably ancillary”. Please evaluate.

11. Correct contributors’ statement and references according to MDPI style.

Overall recommendation: major revision.

Author Response

  1. The manuscript requires extensive editing of English language and style.

Author answer: Thank you. We have done it through Grammarly premium.

  1. Abstract should be improved, especially background and conclusion.

Author answer: glad for this comment; we improved the abstract as suggested.

  1. Why do you mention classification by ILAR? You found this claasification unsatisfactory. Either describe this classification in details or remove it. Introduction is the part of work in which you present what is known about the subject and explain why you decided to write this review- avoid sharing your opinions in introduction.

Author answer: thanks for this comment. We do agree that personal opinions shouldn't be part of the introduction. Therefore, I've modified the sentence as follows “According to specific characteristics displayed in the first six months of the disease, the International League of Associations for Rheumatology (ILAR) recognizes seven main categories: oligoarticular JIA, polyarticular rheumatoid factor (RF) negative JIA, poly-articular RF positive JIA, psoriatic arthritis, enthesitis-related arthritis, systemic JIA, undifferentiated JIA [3]; nevertheless, efforts are currently ongoing to develop a new classification of inflammatory arthritis with particular regard to biological phenotypes [4]”.

  1. Check carefully within the literature the current state of knowledge regarding the growth site in mandibular condyles.

Author answer: fascinating remarks. We added the following sentence with relative references “The condyle is the principal center of mandibular growth; the cartilage ossification at this level causes the increase in the vertical height of the ramus; puberty fasters the mandibular growth, which usually ends by the age of 18 years [8, 9]

  1. Add typical characteristics (phenotype) of JIA patients regarding head morphology, including: retrgnathic profile, antegonial notch, symphysis, face.

Author answer: thank you for this suggestion. We added the following sentence “In JIA patients, the term dentofacial deformity refers to the alteration in in facial bone growth, development, and structure due to TMJ arthritis [12]. Micrognathia, retrognathia, and frontal facial asymmetry are the most frequently reported dentofacial deformity in JIA patients, as well as the deviation of the mandible on maximal mouth opening [13-16].“

  1. Lines 51-52. Small effusion in TMJs can NEVER be seen in healthy children. 

Author answer: thanks for this observation. Stoll et al. reported the presence of mild TMJ synovial fluid (0.90 ± 0.22 mm) in 62/122 (51%) non-arthritic children by MRI [Stoll, M.L.; Guleria, S.; Mannion, M.L.; et al. Defining the normal appearance of the temporomandibular joints by magnetic resonance imaging with contrast: a comparative study of children with and without juvenile idiopathic arthritis. Pediatr Rheumatol Online J 2018;16(1):8.]. It can be argued whether this effusion's size would be significant or not. We modified the sentence as follows: “Indeed, a slight effusion (< 1 mm of diameter) and/or contrast enhancement, in the absence of chronic changes, has been reported in healthy children, too [21]”

  1. PRISMA flow diagram is not visible.

Author answer: Glad for this remark. PRISMA flow chart is now visible.

  1. What does the term „condylar changes” mean? What Kind of condylar changes were the authors looking for?

Author answer: thanks for the opportunity to make this clear. “Condylar changes” refers to every abnormality of mandibular condyle documented by US in JIA patients. We reported the specific condylar alteration for each cited study in this paragraph.

  1. Explain what power Doppler signal is and what information it gives for the clinicians.

Author answer: Sharable comments, thanks. We added the following sentence “Ultrasound can document synovial vascularization and hyperemia through a pow-er Doppler signal; this amplifies the US sensitivity for acute inflammation in several joints [45]”.  

  1. You mentioned that US is indubitably ancillary to MRI. In fact, you did not mention what additional information we get from US which we do not get from MRI. If there is nothing superior, US cannot be called: „indubitably ancillary”. Please evaluate.

Author answer: thanks for this remark. We replaced “ancillary” with “secondary”.

  1. Correct contributors’ statement and references according to MDPI style.

Author answer: thanks for this remark. We modified it accordingly.

Reviewer 3 Report

Dear author,
The article is well written and for this reason i suggest only minor revisions, nevertheless i invite to
 improve the paper.

First, i ask you to check the plagiarism of your article using specific sites to get a similitary report 

You need to review the grammar and English of your article, with the help of a native English speaker (you can specify who helped you in reviewing English in the acknowledgements) or simply by using a site that can support you in English (in this case lease attach after this check a self-certification of English revision).

this may be useful as a basis:

Dear ……., 
as corresponding author, I declare to have extensively revised the manuscript entitled "…….." for English issues, according to the following parameters:
- Formal Language 

- Grammar (correct use of tenses)
- Spellings
- Punctuation/prepositions/articles and typographical errors - No vague content (Clarity of expressions)
- Correct presentation of ideas, facts and logic. 
The professional revision was made through Grammarly premium (https://www.grammarly.com/premium). An official certificate is not available, but I certify to meet all the requirements. 

Best regards, 

I suggest you to add an image in order to improve the iconography of the article and also I suggest you add a table with the list of abbreviations used in the text.

Please be sure to use only keywords accordingly to medical subject headings (Mesh word) for a better indexing.

Moreover I sugget to add  more recent references about the topic of the article, dwelling in the introduction on articles published in 2022 and describing what your article will add compared to the last articles published; Preferably a published articles should be with 90 or more references, you don’t be afraid to add too many references.

I suggest you some articles that will help you improve your article about TMD, and about congental syndromes with orofacial consequences similar to the Juvenile Idiopathic Arthritis (JIA).
Teledentistry in the Management of Patients with Dental and Temporomandibular Disorders Doi: https://doi.org/10.1155/2022/7091153 
Prosthodontic Treatment in Patients with Temporomandibular Disorders and Orofacial Pain and/or Bruxism: A Review of the Literature https://doi.org/ 10.3390/prosthesis4020025 
Oral-facial-digital syndrome (OFD): 31-year follow-up management and monitoring PMID: 29460530 

Telescopic overdenture on natural teeth: Prosthetic rehabilitation on OFD syndromic patient and a review on available literature PubMed ID 29460531

Please expand conclusion section with main results and future perspectives of this study.

Remember for your future review that is fundamental  register the protocol on Prospero site:

https://www.crd.york.ac.uk/prospero/

Thank You,

Kind Regards

Author Response

You need to review the grammar and English of your article, with the help of a native English speaker (you can specify who helped you in reviewing English in the acknowledgements) or simply by using a site that can support you in English (in this case lease attach after this check a self-certification of English revision).

this may be useful as a basis:

Author answer: thanks for the suggestion, we reviewed the manuscript through Grammarly premium as suggested. Here is the self-certification of English revision:

Dear Editors of Children, 
as corresponding author, I declare to have extensively revised the manuscript entitled "
Role of ultrasound evaluation of temporomandibular joint in juvenile idiopathic arthritis: a systematic review." for English issues, according to the following parameters:
- Formal Language 
- Grammar (correct use of tenses)
- Spellings
- Punctuation/prepositions/articles and typographical errors - No vague content (Clarity of expressions)
- Correct presentation of ideas, facts and logic. 
The professional revision was made through Grammarly premium (https://www.grammarly.com/premium). An official certificate is not available, but I certify to meet all the requirements. 

I suggest you to add an image in order to improve the iconography of the article and also I suggest you add a table with the list of abbreviations used in the text.

Author answer: glad for these remarks. As requested, we added the abbreviations list and a US image of TMJ (Figure 1).

Please be sure to use only keywords accordingly to medical subject headings (Mesh word) for a better indexing.

Author answer: we modified the keywords with Mesh terms only.

Moreover I sugget to add  more recent references about the topic of the article, dwelling in the introduction on articles published in 2022 and describing what your article will add compared to the last articles published; Preferably a published articles should be with 90 or more references, you don’t be afraid to add too many references.

Author answer: thanks for the suggestion. We added up-to-date references.

Please expand conclusion section with main results and future perspectives of this study.

Author answer: glad for this comment, we have done what was requested.

Remember for your future review that is fundamental register the protocol on Prospero site:

https://www.crd.york.ac.uk/prospero/

Author answer: sure, thanks.

Round 2

Reviewer 2 Report

I accept the submitted changes.

Kind regards.